# Obstructive sleep apnea in obese pregnant women: A prospective study

L. Ghesquière[1,2]*, P. Deruelle[1,2], Y. Ramdane[3], C. Garabedian[1,2], C. Charley-Monaca[4], A.-F. Dalmas[5]

1 EA 4489 –Perinatal Environment and Health, University of Lille, Lille, France, 2 Department of Obstetrics, CHU Lille, Lille, France, 3 Department of Biostatistics, EA 2694 –Public Health: Epidemiology and Quality of Care, University of Lille, CHU Lille, Lille, France, 4 Department of Clinical Neurophysiology–Sleep Disorders Unit, University of Lille, CHU Lille, Lille, France, 5 Department of Anesthesia–Intensive Care, University of Lille, Lille, France

* louise.ghesquiere88@gmail.com

## Abstract

### Objective

Define the prevalence of OSA in a population of obese pregnant women. Secondary objectives were to assess its obstetric consequences and define its risk factors in this population.

### Methods

This single-center prospective study took place at the Lille University Hospital from 2010 to 2016 and included pregnant women with a body mass index (BMI) > 35 kg/m$^2$. They underwent polysomnography (type 1 sleep testing) between 24 and 32 weeks of gestation to diagnose OSA. Clinical, obstetric, and fetal data were collected monthly and at delivery. We compared the groups with and without OSA and calculated its prevalence.

### Results

This study included 67 women with a mean BMI of 42.4 ± 6.2 kg/m$^2$. Among them, 29 had OSA, for a prevalence of 43.3% (95% confidence interval, 31.4–55.2); it was mild or moderate in 25 women and severe in 4. Comparison of the two groups showed that women in the OSA group were older (31.9 ± 4.7 years vs 29.5 ± 4.8 years, $P$ = .045), had chronic hypertension more frequently (37.9% vs 7.9%, $P$ = .0027), and had a higher mean BMI (43.8 ± 6.2 kg/m$^2$ vs 41.2 ± 6 kg/m$^2$, $P$ = .045). During pregnancy, they developed gestational diabetes more often (48.3% vs 23.7%, $P$ = .04). No significant differences were observed for any of the other criteria studied.

### Conclusions

The prevalence of OSA was high in our study, and women with it developed gestational diabetes during pregnancy more often. No other obstetric complications were observed.

**Data Availability Statement:** The study's minimal underlying data set is within the Supporting Information files.

**Funding:** The financial support provided by the public state amounted to 10,000 euros. This support was used to finance polysomnography equipment, the cost of patient nights in hospital (572.94 euros in 2007, 551.74 euros in 2008, 554.50 euros since 2009), medical and paramedical staff, and a clinical researcher. The funder had no role in study design, data collection and analysis to publish or preparation of the manuscript.

**Competing interests:** The authors have declared that no competing interests exist.

## Introduction

Obstructive sleep apnea (OSA) is a high-prevalence disease, sometimes exceeding 50% in the general population [1–4]. The intermittent hypoxia and fragmentation of sleep it engenders are risk factors for cardiovascular diseases, especially for chronic hypertension, metabolic syndrome, and diabetes [2–5]. Clinically, this disorder is manifested principally by two symptoms: daytime somnolence and nocturnal snoring. Polysomnography in a sleep laboratory is the reference examination for this diagnosis, defined by calculating the apnea-hypopnea index (AHI). In Western countries, the prevalence of mild OSA (AHI $\geq$ 5) has been estimated at 9–38% and that of moderate to severe OSA (AHI $\geq$ 15) at 6–17% [2]. These variations are explained by differences in the diagnostic criteria, but also by the heterogeneity of study populations; prevalence is higher among men, the elderly, and obese women. Obesity is its principal risk factor.

Obesity in pregnant women is accompanied by an increase in pregnancy-related vascular complications such as preeclampsia, pregnancy-related hypertension, and gestational diabetes [6, 7]. Other obstetric complications are associated with an impaired quality of labor (higher rate of post-term pregnancies, prolonged labor, and cesareans for cervical dystocia) [8–13] and a higher risk of postpartum hemorrhage in vaginal deliveries [9]. Obesity in pregnant women may also be a risk factor for the development of sleep apnea, which may further increase the risk of pregnancy complications.

OSA during pregnancy has been studied often. Depending on the definition used and the study, its prevalence among women of child-bearing age has been estimated at 1.4–16.9% [2, 14, 15]. But the exact prevalence among pregnant women remains unknown, especially because it is underestimated and underdiagnosed in this population because of its nonspecific clinical symptoms during pregnancy (asthenia, nonrestorative sleep, snoring in the third trimester) that may thus be trivialized by both women and clinicians [16, 17]. Moreover, because many of the studies about OSA and pregnancy have not used polysomnography, it may well have been either under- or overdiagnosed.

Substantially less is known about the effects of OSA in pregnant women than in nonpregnant populations. Recent data indicate it is associated with higher risks of gestational diabetes, preeclampsia, and fetal growth restriction (FGR). A meta-analysis published in 2018 showed that women with OSA are also at higher risk of preterm, cesarean, and operative vaginal deliveries, as well as of postoperative complications [18]. Nonetheless, most of the data currently available is limited to case reports or studies without either or both of an appropriate, objective test to diagnosis OSA and adjustment for obesity, an obvious confounding factor [19, 20].

Few studies have specifically explored OSA in pregnant women with obesity. We therefore chose to conduct a study in this population: its principal objective was to define their prevalence of OSA. Our hypothesis was that its prevalence would be higher among them than among non-obese women. Our secondary objectives were to compare the women with and without OSA for the course and outcomes of their pregnancy and to identify some of its predictive factors.

## Material and methods

This prospective single-center study took place at the Lille University Hospital Center, at the Jeanne de Flandre maternity ward, and included a population of obese pregnant women who were offered overnight in-hospital polysomnography for OSA screening. The Lille Hospital Ethics Committee approved this study (CPP 09/65 N° 2009-A01018-49).

Its principal objective was to study the prevalence of OSA in the population of obese women receiving prenatal care and giving birth at our hospital, specifically those with a

prepregnancy BMI >35 kg/m$^2$. The first of our secondary objectives was to compare the course of pregnancy in women with and without OSA, especially for the onset of pregnancy-related vascular disorders (pregnancy-related hypertension, preeclampsia, eclampsia, HELLP syndrome, or fetal growth restriction (FGR), defined by birthweight <10th percentile for gestational age), but also for their pregnancy outcome (type of delivery and newborn's characteristics). The last secondary objective was to examine whether various criteria, including maternal age, parity, BMI, history of chronic hypertension, history of diabetes, family history of OSA, weight gain during pregnancy, and gestational diabetes might be risk factors for developing OSA in this population.

Participation in the study was offered to all pregnant women aged at least 18 years receiving prenatal care at the Jeanne de Flandre maternity ward with a prepregnancy BMI >35 kg/m$^2$. The polysomnography had to take place after 24 and before 32 weeks of gestation. Women were excluded if they refused to participate, did not sign the informed consent, had a twin or higher-order multiple pregnancy, had a guardian or conservator, or took medication likely to modify OSA (antipsychotics, anxiolytics, hypnotics and sedatives and muscle relaxants benzodiazepine). Obstetricians explained the study to women meeting the inclusion criteria in a special consultation and gave them written information about it. After time to consider participation, the women who agreed provided written informed consent before inclusion.

These women were then followed up monthly from the fourth month of pregnancy, during prenatal consultations, when obstetric data were collected. Fetal monitoring took place by monthly ultrasound to evaluate fetal weight and by fetal and uterine artery Doppler scans. Polysomnography took place between 24 and 32 weeks of gestation. Mode of delivery and neonatal status were recorded at birth.

The following obstetric data were collected during pregnancy: maternal age, parity, prepregnancy BMI, type 1 or 2 diabetes, chronic hypertension, history of phlebitis and/or pulmonary embolism, family member with OSA, gestational diabetes, weight gain during pregnancy, hospitalization during pregnancy, and presence of any pregnancy-related vascular disease or complication (pregnancy-related hypertension, preeclampsia, eclampsia, HELLP syndrome, or FGR). These pregnancy-related vascular diseases were grouped together as a composite criterion. The criteria for their diagnosis were those defined by the French national guidelines [21, 22].

Polysomnography was performed during one night of hospitalization in the hospital's sleep laboratory, recording an electroencephalogram, electro-oculograms, submental and bilateral anterior tibialis electromyography, an electrocardiogram, nasal and oral air-flow, oxygen saturation and thoracic and abdominal movement. Various clinical indicators were also collected to assess the existence of OSA (ronchopathy or snoring, respiratory pauses, nocturia, night sweats, morning headaches, perception of nonrestorative sleep, and excessive daytime somnolence). Analysis of these clinical and polysomnographic data enabled us to define two groups of women: one group with OSA and one without it. AHI ≥5 measured by polysomnography defined OSA. Among the women with OSA, we distinguished those with mild or moderate sleep apnea (AHI <30) and those with severe OSA (AHI ≥30). Severity was also assessed by simultaneous measurement of arterial oxygen desaturation and consideration of the complete clinical picture (hypersomnolence, neuropsychological disorders, and hypertension). Ventilation by continuous positive airway pressure (CPAP) was proposed to the women with severe OSA, with continuing medical follow-up at the sleep center.

Data about delivery and the baby were collected postpartum: mode of delivery, spontaneous or induced labor, term at delivery, birth weight, acid-base status, and NICU transfer.

The sample size was calculated on the basis of the estimated 95% confidence interval (95% CI) of the theoretical frequency of women with a BMI >35 and OSA (that is, the principal

study objective). This 95% CI was calculated with Sachs' method, and the sample size was calculated to obtain a given level of precision for this CI, defined as half of it. By using the standard formula, which can determine positive predictive value from sensitivity, specificity, and prevalence, we were able to estimate the frequency of OSA in women with a BMI >35 kg/m$^2$ at 56%. We set the precision at 12.5% (length of CI: 25%). In these conditions, we calculated that 68 women were necessary for the study (theoretical frequency estimated at 43.3–68.2%) [23]. The frequency of OSA was estimated by its 95% CI. For the descriptive data analysis, the qualitative data were presented as numbers and percentages, and the quantitative data as means and their standard deviations. The normality of the numeric parameters was verified graphically and by the Shapiro-Wilk test. The groups with and without OSA were compared by Chi-square or Fisher's exact tests for the qualitative variables (e.g., age, BMI, weight gain, birth weight, Apgar score) and by the Kruskal Wallis or Mann-Whitney tests for the quantitative variables (e.g., chronic hypertension, diabetes, the vascular disease composite criterion). Significance was set at 5%. The analysis was performed with SAS software (version 9.4, SAS Institute, Cary, NC).

## Results

In all, 86 pregnant women with a BMI >35 met the inclusion criteria; 19 were finally excluded because they had not undergone polysomnography (Fig 1). This statistical analysis included 67 patients. The characteristics of the excluded women did not differ significantly from those of the others.

Table 1 presents the 67 women's characteristics: their mean age was 30.5 years ± 4.9, and their mean BMI 42.4 ± 6.2. Fourteen (20.9%) had a history of chronic hypertension, 14 (20.9%) of diabetes and 6 (9%) of the two.

Our principal objective was to assess the prevalence of OSA in our population. Among these 67 pregnant women, 29 (43.3%, 95% CI, 31.4–55.2) had OSA, 25 of them mild or moderate and 4 severe (Fig 1). These women's median AHI was 10.9 (9.2–19.7) and their median

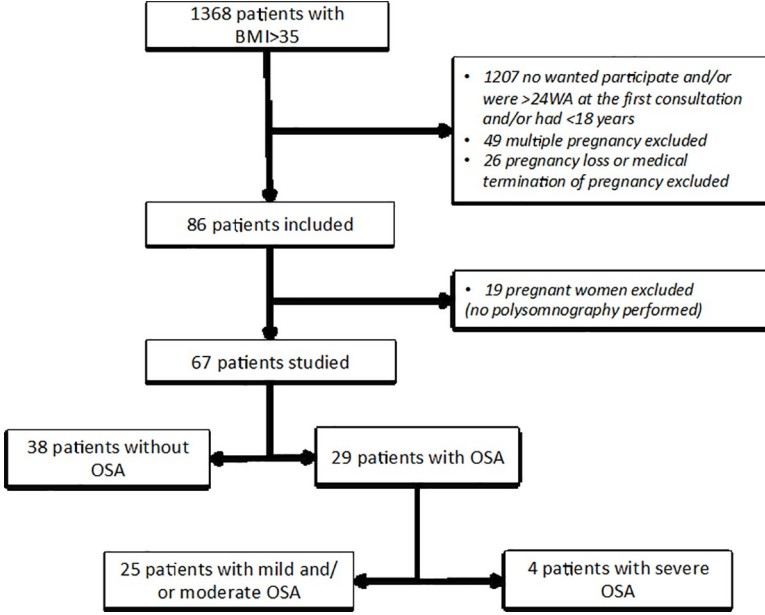

**Fig 1. Flow chart.** BMI = Body Mass Index; OSA = obstructive sleep apnea.

**Table 1. Maternal characteristics for all patients and compared to OSA group and no OSA group to study risk factors for developing an OSA.**

| Maternal characteristics | All patients (N = 67) | OSA group (N = 29) | No OSA group (N = 38) | p |
|---|---|---|---|---|
| Age (years) | 30.5 ± 4.9 | 31.9 ± 4.7 | 29.5 ± 4.8 | **0.04** |
| Parity | 1.1 ± 1.5 | 1.2 ± 1.5 | 1.1 ± 1.5 | 0.52 |
| BMI (kg/m$^2$) | 42.4 ± 6.2 | 43.8 ± 6.2 | 41.2 ± 6 | **0.05** |
| Diabetes (type 1or type 2) | 14 (20.9) | 8 (27.6) | 6 (15.8) | 0.24 |
| Chronic hypertension | 14 (20.9) | 11 (37.9) | 3 (7.9) | **0.003** |
| Family member with OSA | 16 (23.9) | 7 (24.1) | 9 (23.7) | 0.97 |
| Gestational diabetes | 23 (34.3) | 14 (48.3) | 9 (23.7) | **0.04** |
| Weight gain (kg) | 6.3 ± 7.9 | 7.6 ± 7.9 | 5.4 ± 8 | 0.31 |

BMI = Body Mass Index; OSA = obstructive sleep apnea.

Data are on average ± standard deviation or N (%). p < 0.05 is significant.

time with oxygen saturation <90% 4 minutes (1–13 minutes). Of the four women with severe OSA, only three agreed to start CPAP treatment and only one continued it to the end of her pregnancy.

The first of our secondary objectives was to compare the OSA and non-OSA populations for several criteria (Table 2). During pregnancy, women with OSA developed gestational diabetes more often: 48.3% (n = 14) compared with 23.7% in the non-OSA group (n = 9) (P = 0.04). There were no significant differences for weight gain, hospitalization during pregnancy, or the vascular disease composite criterion, although we note there was a trend toward

**Table 2. Characteristics of pregnancy, delivery and new born according to the OSA or no OSA groups.**

| | | OSA group (N = 29) | No OSA group (N = 38) | p |
|---|---|---|---|---|
| Characteristics of pregnancy: | | | | |
| | Gestational diabetes | 14 (48.3) | 9 (23.7) | **0.04** |
| | Weight gain (kg) | 7.6 ± 7.9 | 5.4 ± 8 | 0.31 |
| | Hospitalization | 21 (72.4) | 25 (67.6) | 0.67 |
| | Vascular complication or disease (composite criterion) | 10 (34.5) | 6 (16.2) | 0.09 |
| Characteristics of delivery: | | | | |
| | Term at delivery (WA) | 38.1 ± 2.9 | 39 ± 2.1 | 0.36 |
| | Preterm birth before 37 WA | 6 (20.7) | 5 (13.2) | 0.41 |
| | Preterm birth before 32 WA | 2 (6.7) | 0 (0) | 0.10 |
| | Vaginal delivery | 13 (46.4) | 24 (64.9) | 0.14 |
| | Cesarean section | 16 (55.2) | 13 (35.1) | 0.10 |
| | Scheduled cesarean section | 7 (24.1) | 3 (8.3) | 0.10 |
| | Cesarean section during labor | 5 (17.2) | 9 (25) | 0.45 |
| | Emergency cesarean section | 4 (13.8) | 2 (5.6) | 0.39 |
| | Induced labor | 16/28 (57.1) | 13 (35.1) | 0.08 |
| Neonatal characteristics: | | | | |
| | Birth weight (kg) | 3165 ± 937.5 | 3292 ± 644.7 | 0.98 |
| | pH status | 7.24 ± 0.10 | 7.24 ± 0.09 | 0.99 |
| | NICU transfer | 4 (13.8) | 3 (8.3) | 0.69 |

BMI = Body Mass Index; OSA = obstructive sleep apnea; WA = weeks of amenorrhea.

Data are on average ± standard deviation or N (%). p < 0.05 is significant

more complications in the OSA group (n = 10 vs n = 6, 34.5% vs 16.2%, *P* = 0.09). Among these patients who had vascular disease complications, 4 had a history of chronic hypertension (28.6%) and 2 of diabetes (13.3%). The groups did not differ significantly about characteristics of delivery but there was a trend for more induction (n = 16 (57.1%) vs n = 13 (35.1%), P = 0.08) and more cesarean section (n = 16 (55.2%) vs n = 13 (35.1%), P = 0.10) in OSA group. Among patients who had cesarean sections, 10/16 (62.5%) in OSA group and 5/13 (38.5%) in non OSA group had a history of chronic hypertension and/or diabetes. Among patients who had induction of labor, 10/16 (62.5%) in OSA group and 6/13 (46.2%) in non OSA group had a history of chronic hypertension and/or diabetes. Similarly, there were no significant differences for neonatal characteristics: birth weight was 3165 g ± 937.5 in the OSA group and 3292 g ± 644.7 in the non-OSA group (*P* = 0.98) with respectively, a pH of 7.24 ± 0.10 and 7.24 ± 0.09 (*P* = 0.99) and 4 (13.8%) and 3 (8.3%) NICU admissions (*P* = 0.69).

For the second of our secondary objectives, significant difference was observed between the groups for women's age, prepregnancy history of hypertension, gestational diabetes, and mean BMI (Table 1). The mean age of the women in the OSA group was 31.9 ± 4.7 years vs 29.5 ± 4.8 years (*P* = 0.045) in the women with this sleep problem, and their mean BMI respectively 43.8 ± 6.2 kg/m$^2$ and 41.2 ± 6 kg/m$^2$ (*P* = 0.045). Prepregnancy hypertension was found in 11 women with OSA (37.9%) and only 3 without it (7.9%) (*P* = 0.0027). Gestational diabetes occurred among 48.3% of the women with OSA (n = 14) and 23.7% in those without it (n = 9, *P* = 0.036). We observed no significant differences between the groups (*P*>0.05) for the other criteria studied: parity, prepregnancy diabetes, family history of OAS, and weight gain during pregnancy.

## Discussion

The principal objective of our study was to evaluate the prevalence of OSA in a population of pregnant women with a BMI >35 kg/m$^2$. In this high-BMI population at risk, we found a prevalence of 43.3% (n = 29) with OSA mild or moderate in 25 women and severe in 4. The women with OSA developed gestational diabetes during pregnancy more often than the others. They were older and had more maternal comorbidities before pregnancy, a higher BMI (43.8 ± 6.2 kg/m$^2$), and more frequent chronic hypertension.

Prevalence was high in our study. The prevalence of OSA in a general population of pregnant women, regardless of BMI, is not known. It has been estimated, depending on the study, at 1.4–16.9% among women of child-bearing age [2, 14, 15]. In studies of populations of pregnant women at risk of OSA, its prevalence was higher, as in our study. Rice et al. sought to compare the risk of OSA as a function of BMI in a population of pregnant women [24]. They found that the odds ratio for risk of OSA was 3.69 (95% CI; 1.82–7.50) for overweight women (BMI 25–29.9 kg/m$^2$) and 13.23 (95% CI, 6.25–28.01) for obese women (BMI ≥ 30 kg/m$^2$), compared with normal-weight women (BMI <25 kg/m$^2$). Louis et al. studied 175 pregnant women with BMI ≥30 kg/m$^2$ and found a 15.4% prevalence of OSA (13 mild, 9 moderate, and 5 severe) [25]. Their prevalence was lower than ours, perhaps explained by their BMI inclusion criterion (BMI >30 kg/m$^2$), also lower than ours. In addition, although the polysomnographic criteria for an OSA diagnosis were similar to those we used, the term of pregnancy at the time of their testing and diagnosis was not specified; this difference too could explain the difference in prevalence between their results and ours; prevalence of OSA increases with the term of pregnancy. Work by Facco et al. in a population of pregnant women at risk of OSA (BMI ≥30 or chronic hypertension) showed that in the first trimester of pregnancy, the prevalence rates of mild, moderate, and severe OSA were respectively 21%, 6%, and 3%, while in the third trimester, they were 35%, 7%, and 5% (*P* < .001) [26]. These two studies are consistent with our

results in finding a higher prevalence of OSA in a population at risk, especially due to obesity, for mild and moderate cases.

We did not find significantly different pregnancy, delivery, and neonatal complications between the two groups except that gestational diabetes was more frequent in women with this sleep disorder. Our results are not in complete accord with the literature. In 2012, Chen et al. conducted their randomized study of 791 pregnant women with and 3955 without OSA, and found significantly higher risks of preeclampsia, FGR, cesarean delivery, preterm delivery, and low birth weight in the OSA group [27]. The principal limitation of their study was the absence of adjustment for BMI in their Taiwanese population at risk of obesity. In 2014, a meta-analysis examining the consequences of OSA on pregnancy that did adjust for BMI found a risk of developing pregnancy-related hypertension and/or preeclampsia that was 2.34 (95% CI, 1.60–3.09) times higher in women with compared to without OSA, and a risk of gestational diabetes 1.86 (95% CI, 1.30–2.42) times higher [28]. This study, like ours, found no increase in the risk of low birth weight. Another more recent meta-analysis in 2018 focused specifically on the consequences of OSA on delivery and on neonatal condition [18]. After adjustment for age and BMI, this study found increased risks of cesarean and preterm delivery (<37 weeks) as well as of FGR (<2500 g). FGR may bias these results, as it is known to increase the risks of induced preterm birth and of cesarean delivery. These different studies did not target populations of pregnant women at risk of OSA because of high BMI. In their population of obese pregnant women, Louis et al. showed that the women in the OSA group had a higher risk of preeclampsia, cesarean delivery, and newborn transfer to the NICU [25]. After adjustment for various criteria, and especially for BMI, they found that only the risk of preeclampsia was significantly higher. Our study observed that the rate of the composite criterion (of pregnancy-related vascular diseases) was higher in the OSA group, but not significantly so (34.5% vs 16.2%, $P$ = .086). We observed too a trend to more induction and more cesarean section in OSA group, but in these patients, there seemed to have more comorbities. Contrary to our study, theirs found no difference between the groups for gestational diabetes.

Our results about the risk factors associated with OSA are consistent with those of the literature. Chronic hypertension, diabetes, high BMI, and age were all shown to be comorbidities associated with OSA in a general population of obese women [29, 30]. In a population of pregnant women with obesity, Louis et al. also showed that those with OSA had a higher BMI (46.8 ±12.2 vs 38.1± 7.5 kg/m$^2$, $P$ = .002) and more frequent chronic hypertension (55.6 vs 32.4%, $P$ = .02) [25]. The 2018 meta-analysis also reported that the women with OSA were older (RR 1.66, 95% CI, 1.04–2.228) and had a higher BMI (RR 3.31, 95% CI 2.30–4.32) [18].

The principal strength of our study was that OSA was diagnosed objectively by inpatient polysomnography, the reference examination, so that the diagnosis should not be either under- or overestimated. It nonetheless has some limitations. As our population was limited to obese women with BMI >40 kg/m$^2$, we could not calculate the prevalence and extent of OSA in a population of non-obese woman. Even though our population specifically included especially obese women, comparison of the two groups found a significant difference in their BMI: it was higher in the group with OSA (43.8 ± 6.2 kg/m$^2$ vs 41.2 ± 6 kg/m$^2$ for the group without, $P$ = 0.045). We did not adjust our results for BMI, which is an important confounding factor in studies of OSA and pregnancy [27]. A high BMI by itself causes vascular complications, gestational diabetes, and both cesarean and operative vaginal deliveries [6, 7]. We also found in our population the usual comorbidities expected to be associated with morbid obesity (34.3% gestational diabetes, 23.9% vascular complications, 43.3% cesareans). Despite this confounding factor, we did not find a significant difference between the groups for other obstetrical data, such as pregnancy-related vascular diseases or outcomes of either the pregnancy or the infant. The only significant difference shown was a higher frequency of gestational diabetes in the

OSA group. Another limitation was associated with OSA severity. Most cases were mild or moderate, with only four severe enough to merit CPAP. We therefore could not study its benefits for these outcomes. Finally, our observation of trends that did not reach significance for some of the criteria studied (the composite criterion, cesarean deliveries, and labor induction) suggests a lack of power for the secondary outcomes due to an insufficient number of women included.

Is screening for OSA during pregnancy useful? We found, as in the literature, many cases of mild or moderate OSA, for which CPAP treatment is not recommended. Moreover, the effectiveness of CPAP has not been assessed in pregnant women, and supplementary studies are needed. We did not find that women with OSA had high rates of complications during pregnancy and delivery, except for the gestational diabetes that is very frequent in obese women. Moreover, diagnosing OSA imposed the constraint of a night of hospitalization and had nontrivial financial costs. It therefore appears important to target women at risk before proposing OSA screening and to assess the real impact of this screening on the course and outcomes of pregnancy.

## Conclusion

Obese pregnant women (BMI >35 kg/m$^2$) are at risk of OSA, highly prevalent during pregnancy. Obstetric complications associated with OSA during pregnancy have been described but studies to define them more precisely remain necessary. The utility of treatment by CPAP during pregnancy has still not been studied. Management of obesity by lifestyle and dietary changes can be tried in these women preventively before conception, especially if other comorbidities are present (older age, chronic hypertension) to reduce the risk of OSA and of the obstetric complications that can be associated with it.

## Supporting information

**S1 Data.**
(ZIP)

## Acknowledgments

We thank all the people who contributed to this study, in particular the clinical researchers, the midwives and the medical staff of the clinical investigation center.

## Author Contributions

**Conceptualization:** P. Deruelle, C. Charley-Monaca, A.-F. Dalmas.

**Data curation:** Y. Ramdane, C. Charley-Monaca, A.-F. Dalmas.

**Formal analysis:** Y. Ramdane, A.-F. Dalmas.

**Investigation:** P. Deruelle, A.-F. Dalmas.

**Methodology:** P. Deruelle, C. Charley-Monaca, A.-F. Dalmas.

**Project administration:** P. Deruelle, A.-F. Dalmas.

**Supervision:** P. Deruelle, C. Garabedian, C. Charley-Monaca, A.-F. Dalmas.

**Validation:** P. Deruelle, Y. Ramdane, C. Garabedian, C. Charley-Monaca, A.-F. Dalmas.

**Visualization:** P. Deruelle, C. Garabedian, C. Charley-Monaca, A.-F. Dalmas.

**Writing – original draft:** A.-F. Dalmas.

**Writing – review & editing:** L. Ghesquière, A.-F. Dalmas.

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
