## [Decision Letter · Decision Letter 0]

16 Jun 2020

PONE-D-20-13731

Obstructive sleep apnea in obese pregnant women: a prospective study

PLOS ONE

Dear Dr. ghesquière,

Thank you for submitting your manuscript to PLOS ONE. After careful consideration, we feel that it has merit but does not fully meet PLOS ONE’s publication criteria as it currently stands. Therefore, we invite you to submit a revised version of the manuscript that addresses the points raised during the review process.

SPECIFIC ACADEMIC EDITOR COMMENTS: There were two expert reviewers in the field that handled your manuscript. We greatly thank them for their time. Although interest was found in your study, there were several major concerns that arose during review. These concerns include comments about definitions of morbidities in the sample population and questions about the statistical analysis. Please address all comments from the reviewers in your revised manuscript.

We look forward to receiving your revised manuscript.

Kind regards,

Frank T. Spradley

Academic Editor

PLOS ONE

"The funders had no role in study design, data collection and analysis, decision to publish, or preparation of the manuscript"

Reviewers' comments:

Reviewer's Responses to Questions

**Comments to the Author**

1. Is the manuscript technically sound, and do the data support the conclusions?

Reviewer #1: Yes

Reviewer #2: Yes

2. Has the statistical analysis been performed appropriately and rigorously? 

Reviewer #1: Yes

Reviewer #2: Yes

3. Have the authors made all data underlying the findings in their manuscript fully available?

Reviewer #1: Yes

Reviewer #2: Yes

4. Is the manuscript presented in an intelligible fashion and written in standard English?

Reviewer #1: Yes

Reviewer #2: Yes

5. Review Comments to the Author

Reviewer #1: The manuscript “Obstructive sleep apnea in obese pregnant women: a prospective study” by Ghesquière et al. included analysis of 67 obese pregnant women, with and without OSA. The authors report the prevalence of OSA in those women, as well as clinical morbidities such as diabetes, hypertension, and effects on the babies. The study is very nicely done, and the results are presented clearly. The authors did a very nice job of addressing other studies with results that differed from their own, as well as pointing out the caveats in their own study. This information is important, and in general, is lacking in the field. The attention to BMI here is an important variable that is usually not well-addressed.

I have just a few minor issues that if addressed, would help to strengthen this manuscript.

1) It would be interesting to and important for others in the field if the authors could indicate which morbidities were present in the same women. For example, was it the same women who had diabetes and chronic hypertension? And was it those women who had vascular complications and had to have C-sections and/or induced labor? This information is important to better understand the consequences of OSA on these outcomes.

2) In a similar vein, there appear to be statistical trends in some of the endpoints listed, especially in Table 2, that with additional power, would likely be able to be sussed out. In the Results, one of the "statistical trends" was discussed, but others were not. Often, p values 

3) While the paper is nicely written, there is too much repetition and redundancy throughout. For example, in the Methods, when discussing the gestational diabetes in the Results section on p8, and the 4 women with severe OSA who were offered CPAP treatment.

Reviewer #2: In the article under review the authors present a prospective study where they performed inpatient polysomnography sleep testing between 24 and 32 weeks gestation on a cohort of obese pregnant women all with a BMI of >35. They then compared groups with and without OSA and looked at birth outcomes associated with OSA in a cohort of women at a University Hospital in Lille France. The study included a total of 67 women.

Although the authors did perform a power calculation, prior large prospective studies determined sample size with the primary objective of a tool to screen obese pregnant women for OSA. They assumed a 60% detection rate and estimated they needed 182 subjects. I am wondering why the calculation differed so much in this current study with almost half the participants needed? The authors were also underpowered to look at secondary outcomes and should note that. This is an interesting study that will be an important contribution to the literature and is well written.

Specific comments include:

1) Please review how chronic hypertension was defined as well as the French definitions for preeclampsia

2) Please define what medications were part of the exclusion criteria

3) The authors should state in the limitations that they were underpowered for the secondary outcomes.

4) FGR as defined as <2500 grams does not account for gestational age. The authors should use either SGA (birthweight <10th percentile for gestational age) or the appropriate in utero estimate

5) The authors should consider removing Table 1 and presenting the clinical and demographic factors by study group (OSA versus no OSA) as well as the outcomes by study group

6) The authors should show a flow sheet with their entire population (all women BMI >35) and then exclusion criteria, pregnancy loss, withdrew, delivered elsewhere, did not participate, stillbirths? Periviable deliveries?

7) Table 3 should be table 1 (maternal characteristics)

8) Is there information on race/ethnicity? Smoking? Asthma?

9) Can the authors report gestational weeks of delivery? Preterm birth before 37 weeks? 32 weeks?

6. PLOS authors have the option to publish the peer review history of their article (what does this mean?). If published, this will include your full peer review and any attached files.

Reviewer #1: No

Reviewer #2: No

---

## [Author Response · Author response to Decision Letter 0]

13 Aug 2020

Reviewers' comments:

Reviewer's Responses to Questions

Comments to the Author

1. Is the manuscript technically sound, and do the data support the conclusions?

Reviewer #1: Yes

Reviewer #2: Yes

2. Has the statistical analysis been performed appropriately and rigorously? 

Reviewer #1: Yes

Reviewer #2: Yes

3. Have the authors made all data underlying the findings in their manuscript fully available?

Reviewer #1: Yes

Reviewer #2: Yes

4. Is the manuscript presented in an intelligible fashion and written in standard English?

Reviewer #1: Yes

Reviewer #2: Yes

5. Review Comments to the Author

Reviewer #1: 

The manuscript “Obstructive sleep apnea in obese pregnant women: a prospective study” by Ghesquière et al. included analysis of 67 obese pregnant women, with and without OSA. The authors report the prevalence of OSA in those women, as well as clinical morbidities such as diabetes, hypertension, and effects on the babies. The study is very nicely done, and the results are presented clearly. The authors did a very nice job of addressing other studies with results that differed from their own, as well as pointing out the caveats in their own study. This information is important, and in general, is lacking in the field. The attention to BMI here is an important variable that is usually not well-addressed.

I have just a few minor issues that if addressed, would help to strengthen this manuscript.

1) It would be interesting to and important for others in the field if the authors could indicate which morbidities were present in the same women. For example, was it the same women who had diabetes and chronic hypertension? And was it those women who had vascular complications and had to have C-sections and/or induced labor? This information is important to better understand the consequences of OSA on these outcomes.

Thank you for this suggestion. I added some results (just descriptive) and one sentence in discussion:

In results: “Fourteen (20.9%) had a history of chronic hypertension, 14 (20.9%) of diabetes and 6 (9%) of the two”

“Among these patients who had vascular disease complications, 4 had a history of chronic hypertension (28.6%) and 2 of diabetes (13.3%).”

“Among patients who had cesarean sections, 10/16 (62.5%) in OSA group and 5/13 (38.5%) in non OSA group had a history of chronic hypertension and/or diabetes. Among patients who had induction of labor, 10/16 (62.5%) in OSA group and 6/13 (46.2%) in non OSA group had a history of chronic hypertension and/or diabetes.”

In discussion: “We observed too a trend to more induction and more cesarean section in OSA group, but in these patients, there seemed to have more comorbities”

2) In a similar vein, there appear to be statistical trends in some of the endpoints listed, especially in Table 2, that with additional power, would likely be able to be sussed out. In the Results, one of the "statistical trends" was discussed, but others were not. Often, p values 

Thank you for your remark. I changed the sentence: “Among the characteristics of delivery, The groups did not differ significantly about characteristics of delivery for rates of vaginal delivery, cesarean delivery, or labor induction (P>0.05) but there was a trend for more induction (n=16 (57.1%) vs n=13 (35.1%), P=0.08) and more cesarean section (n=16 (55.2%) vs n=13 (35.1%), P=0.10) in OSA group”

3) While the paper is nicely written, there is too much repetition and redundancy throughout. For example, in the Methods, when discussing the gestational diabetes in the Results section on p8, and the 4 women with severe OSA who were offered CPAP treatment.

Many thanks for this suggestion, you’re right. I modified and I removed repetitions : 

In the results: The For the second of our secondary objectives, was to study whether various criteria, including age, parity, BMI, history of chronic hypertension, history of diabetes, family history of OSA, weight gain during pregnancy, and gestational diabetes, might be risk factors for developing OSA in this population. A a significant difference was observed between the groups for women's age, prepregnancy history of hypertension, gestational diabetes, and mean BMI (Table 3)

In the discussion: Of the four women with severe OSA, only three agreed to start CPAP treatment and only one continued it to delivery.

Reviewer #2: In the article under review the authors present a prospective study where they performed inpatient polysomnography sleep testing between 24 and 32 weeks gestation on a cohort of obese pregnant women all with a BMI of >35. They then compared groups with and without OSA and looked at birth outcomes associated with OSA in a cohort of women at a University Hospital in Lille France. The study included a total of 67 women.

Although the authors did perform a power calculation, prior large prospective studies determined sample size with the primary objective of a tool to screen obese pregnant women for OSA. They assumed a 60% detection rate and estimated they needed 182 subjects. I am wondering why the calculation differed so much in this current study with almost half the participants needed? The authors were also underpowered to look at secondary outcomes and should note that. This is an interesting study that will be an important contribution to the literature and is well written.

Specific comments include:

1) Please review how chronic hypertension was defined as well as the French definitions for preeclampsia : 

Thank you for this remark. I added a reference (N°22) to the recommendations of the French society of arterial hypertension in the references, which defined chronic arterial hypertension and preeclampsia: 22. Consensus d'Experts de la Société Française d'Hypertension Artérielle (SFTA). HTA et Grossesse [Internet]. 2015. Available from: http://www.sfhta.eu/wp-content/uploads/2017/03/ Consensus-dexperts-HTA-et-Grossesse-de-la-SFHTA-D%C3%A9c.-2015.pdf”

2) Please define what medications were part of the exclusion criteria

I added types of medications: “or took medication likely to modify OSA (antipsychotics, anxiolytics, hypnotics and sedatives and muscle relaxants).”

3) The authors should state in the limitations that they were underpowered for the secondary outcomes.

I added in discussion: “Finally, our observation of trends that did not reach significance for some of the criteria studied (the composite criterion, cesarean deliveries, and labor induction) suggests a lack of power for the secondary outcomes due to an insufficient number of women included.”

4) FGR as defined as <2500 grams does not account for gestational age. The authors should use either SGA (birthweight <10th percentile for gestational age) or the appropriate in utero estimate

Thank you for this remark. In our study, FGR was defined as birthweight < 10th percentile for gestational age. I added in the methods: « defined by birthweight <10th percentile for gestational age)”

5) The authors should consider removing Table 1 and presenting the clinical and demographic factors by study group (OSA versus no OSA) as well as the outcomes by study group

I removed this table and table 3 is became table 1.

Table 1: Maternal characteristics for all patients and compared to OSA group and no OSA group to study risk factors for developing an OSA. 

Maternal characteristics All patients 

(N=67) OSA group (N=29) No OSA group

(N=38) p

Age (years)* 30.5 ± 4.9 31.9 ± 4.7 29.5 ± 4.8 0.04

Parity* 1.1 ± 1.5 1.2 ± 1.5 1.1 ± 1.5 0.52

BMI (kg/m2)* 42.4 ± 6.2 43.8 ± 6.2 41.2 ± 6 0.05

Diabetes (type 1or type 2) ° 14 (20.9) 8 (27.6) 6 (15.8) 0.24

Chronic hypertension ° 14 (20.9) 11 (37.9) 3 (7.9) 0.003

Family member with OSA° 16 (23.9) 7 (24.1) 9 (23.7) 0.97

Gestational diabetes° 23 (34.3) 14 (48.3) 9 (23.7) 0.04

Weight gain (kg)* 6.3 ± 7.9 7.6 ± 7.9 5.4 ± 8 0.31

BMI = Body Mass Index; OSA = obstructive sleep apnea. 

(*) and (°) : Data are on average ± standard deviation* or N (%)°. p < 0.05 is significant. 

6) The authors should show a flow sheet with their entire population (all women BMI >35) and then exclusion criteria, pregnancy loss, withdrew, delivered elsewhere, did not participate, stillbirths? Periviable deliveries?

I modified the figure 1. Lots of patients were at more of 24WA at the first consultation, because majority of our patients have just the last consultations (7 months, 8 months and 9 months’s consultations) at the maternity. It was for that it was difficult to have the necessary number of patients for the study.

Figure 1 : Flow chart :

7) Table 3 should be table 1 (maternal characteristics)

Table 3 became table 1

Table 1: Maternal characteristics for all patients and compared to OSA group and no OSA group to study risk factors for developing an OSA. 

Maternal characteristics All patients 

(N=67) OSA group (N=29) No OSA group

(N=38) p

Age (years)* 30.5 ± 4.9 31.9 ± 4.7 29.5 ± 4.8 0.04

Parity* 1.1 ± 1.5 1.2 ± 1.5 1.1 ± 1.5 0.52

BMI (kg/m2)* 42.4 ± 6.2 43.8 ± 6.2 41.2 ± 6 0.05

Diabetes (type 1or type 2) ° 14 (20.9) 8 (27.6) 6 (15.8) 0.24

Chronic hypertension ° 14 (20.9) 11 (37.9) 3 (7.9) 0.003

Family member with OSA° 16 (23.9) 7 (24.1) 9 (23.7) 0.97

Gestational diabetes° 23 (34.3) 14 (48.3) 9 (23.7) 0.04

Weight gain (kg)* 6.3 ± 7.9 7.6 ± 7.9 5.4 ± 8 0.31

BMI = Body Mass Index; OSA = obstructive sleep apnea. 

(*) and (°) : Data are on average ± standard deviation* or N (%)°. p < 0.05 is significant. 

8) Is there information on race/ethnicity? Smoking? Asthma? 

I don’t have information on race/ethnicity but I have information for smoking and asthma. Just 4 patients smoked and 7 patients had asthma so they did not seem important to us to add this information because of the small number. 

9) Can the authors report gestational weeks of delivery? Preterm birth before 37 weeks? 32 weeks?

I reported in the table 2 and I added preterm birth before 37weeks and before 32weeks. There was no significant difference between 2 groups.

Table 2: Characteristics of pregnancy, delivery and new born according to the OSA or no OSA groups.

 OSA group (N=29) No OSA group

(N=38) p

Characteristics of pregnancy:

 Gestational diabetes° 14 (48.3) 9 (23.7) 0.04

 Weight gain (kg)* 7.6 ± 7.9 5.4 ± 8 0.31

 Hospitalization° 21 (72.4) 25 (67.6) 0.67

 Vascular complication or disease (composite criterion) ° 10 (34.5) 6 (16.2) 0.09

Characteristics of delivery: 

 Term at delivery (WA)* 38.1 ± 2.9 39 ± 2.1 0.36

 Preterm birth before 37 WA 6 (20.7) 5 (13.2) 0.41

 Preterm birth before 32 WA 2 (6.7) 0 (0) 0.10

 Vaginal delivery ° 13 (46.4) 24 (64.9) 0.14

 Cesarean section° 16 (55.2) 13 (35.1) 0.10

 Scheduled cesarean section° 7 (24.1) 3 (8.3) 0.10

 Cesarean section during labor ° 5 (17.2) 9 (25) 0.45

 Emergency cesarean section ° 4 (13.8) 2 (5.6) 0.39

 Induced labor° 16/28 (57.1) 13 (35.1) 0.08

Neonatal characteristics: 

 Birth weight (kg)* 3165 ± 937.5 3292 ± 644.7 0.98

 pH status* 7.24 ± 0.10 7.24 ± 0.09 0.99

 NICU transfer° 4 (13.8) 3 (8.3) 0.69

BMI = Body Mass Index; OSA = obstructive sleep apnea; WA = weeks of amenorrhea. 

(*) and (°) : Data are on average ± standard deviation* or N (%)°. p < 0.05 is significant

---

## [Decision Letter · Decision Letter 1]

24 Aug 2020

Obstructive sleep apnea in obese pregnant women: a prospective study

PONE-D-20-13731R1

Dear Dr. ghesquière,

We’re pleased to inform you that your manuscript has been judged scientifically suitable for publication and will be formally accepted for publication once it meets all outstanding technical requirements.

Kind regards,

Frank T. Spradley

Academic Editor

PLOS ONE

Reviewers' comments:

Reviewer's Responses to Questions

**Comments to the Author**

1. If the authors have adequately addressed your comments raised in a previous round of review and you feel that this manuscript is now acceptable for publication, you may indicate that here to bypass the “Comments to the Author” section, enter your conflict of interest statement in the “Confidential to Editor” section, and submit your "Accept" recommendation.

Reviewer #1: All comments have been addressed

2. Is the manuscript technically sound, and do the data support the conclusions?

Reviewer #1: Yes

3. Has the statistical analysis been performed appropriately and rigorously? 

Reviewer #1: Yes

4. Have the authors made all data underlying the findings in their manuscript fully available?

Reviewer #1: Yes

5. Is the manuscript presented in an intelligible fashion and written in standard English?

Reviewer #1: Yes

6. Review Comments to the Author

Reviewer #1: All my previous concerns have been addressed. This is a very nice manuscript that will be informative to the field.

7. PLOS authors have the option to publish the peer review history of their article (what does this mean?). If published, this will include your full peer review and any attached files.

Reviewer #1: No

---

## [Editor Report · Acceptance letter]

28 Aug 2020

PONE-D-20-13731R1 

Obstructive sleep apnea in obese pregnant women: a prospective study 

Dear Dr. Ghesquière:

I'm pleased to inform you that your manuscript has been deemed suitable for publication in PLOS ONE. Congratulations! Your manuscript is now with our production department. 

Kind regards, 

on behalf of

Dr. Frank T. Spradley 

Academic Editor

PLOS ONE